# Impact of Adaptive Thermogenesis in Mice on the Treatment of Obesity

**DOI:** 10.3390/cells9020316

**Published:** 2020-01-28

**Authors:** Marianela Bastías-Pérez, Sebastián Zagmutt, M Carmen Soler-Vázquez, Dolors Serra, Paula Mera, Laura Herrero

**Affiliations:** 1Department of Biochemistry and Physiology, School of Pharmacy and Food Sciences, Institut de Biomedicina de la Universitat de Barcelona (IBUB), Universitat de Barcelona, E-08028 Barcelona, Spain; 2Centro de Investigación Biomédica en Red de Fisiopatología de la Obesidad y la Nutrición (CIBEROBN), Instituto de Salud Carlos III, E-28029 Madrid, Spain

**Keywords:** obesity, adaptive thermogenesis, brown adipose tissue, basal metabolic rate, thermoneutrality, chronic cold, ambient temperature and body temperature

## Abstract

Obesity and associated metabolic diseases have become a priority area of study due to the exponential increase in their prevalence and the corresponding health and economic impact. In the last decade, brown adipose tissue has become an attractive target to treat obesity. However, environmental variables such as temperature and the dynamics of energy expenditure could influence brown adipose tissue activity. Currently, most metabolic studies are carried out at a room temperature of 21 °C, which is considered a thermoneutral zone for adult humans. However, in mice this chronic cold temperature triggers an increase in their adaptive thermogenesis. In this review, we aim to cover important aspects related to the adaptation of animals to room temperature, the influence of housing and temperature on the development of metabolic phenotypes in experimental mice and their translation to human physiology. Mice studies performed in chronic cold or thermoneutral conditions allow us to better understand underlying physiological mechanisms for successful, reproducible translation into humans in the fight against obesity and metabolic diseases.

## 1. Introduction

Endotherms, such as mammals, are organisms that use the heat released during cell metabolism to maintain a stable internal temperature [1]. This constant central temperature maintenance favors metabolic conditions so that enzymatic reactions can be carried out optimally, allowing endothermic organisms to be active and adapt to various environments through internal thermoregulation [2,3].

Since a stable core temperature is essential for the survival of endotherms, endothermic animals do everything possible to defend their core temperature in colder environments (Figure 1). However, when central temperature defense is not possible, such as during food shortages or seasonal cold periods, many endotherms, including mice, leave homeothermy and engage in seasonal drowsiness or hibernation to conserve energy [4,5].

The experimental mouse, *Mus musculus*, is one of the most commonly used model organisms for studies of metabolism, immunity and cardiovascular physiology, and for modelling human diseases [6,7,8]. The reason is the conservation of genes between mice and humans, along with the growing repertoire of genetic tools that allow the manipulation of mouse genes to decipher mechanisms underlying physiological and pathophysiological processes. Therefore, we assume that research in mice will provide valuable information on human biology. Although this is true in most studies, there is a considerable difference between the physiology of mice and humans that could directly bias the preclinical findings [9].

Like other small mammals, the mouse has a large surface area and a small body mass. This makes mice vulnerable to fluctuations in the ambient temperature (T_a_), especially when it falls below their thermoneutral temperature (29–31 °C) [9,10,11,12,13]. Mammals try to maintain their core temperature through the adaptive capacity of thermoregulation. Thus, the mouse uses various adaptations to keep thermal homeostasis in colder environments. For instance, the function of brown adipose tissue (BAT) is to maintain body temperature through a process called thermogenesis or heat production. Currently, most metabolic studies involving rodents are carried out at 21 °C, which is a thermoneutral zone in adult humans but is below the thermoneutral zone in mice. As a consequence, research studies in mice that are housed at 21 °C may not directly apply to humans, who live mainly in their comfort zone or neutrality [6,7,10]. For this reason, it is necessary to understand how T_a_ affects metabolic and cardiovascular phenotypes in mice, and the importance of this variable in the modelling of human diseases in rodents.

As animal models and measurement techniques become increasingly accurate and sophisticated, environmental variables become critical for research development. A stable, defined environment is essential to generate consistent experimental results that support both replication and valid interpretations of the data. Previous studies have shown how mice adaptation to T_a_ alters their disease phenotype [14,15,16]. Consequently, T_a_ might be a variable to consider in metabolic studies to guarantee valid interpretation of experimental results, consistent conclusions and greater certainty in the translation of preclinical experiments to clinical studies.

The prevention and treatment of obesity has become a health priority. There is an alarming increase in the prevalence of obesity and associated metabolic diseases, including type 2 diabetes mellitus (T2D) and cardiovascular disease [17]. The health and economic impact of monitoring and managing obesity and associated complications is also remarkable. Lifestyle changes, such as dietary interventions and/or increased physical activity, have been widely recommended to prevent and treat obesity. However, it is essential to determine why, in general, many obese individuals are exceptionally resistant to treatment and voluntary weight loss is so difficult to achieve and sustain over time. Thus, a better understanding of energy homeostasis is essential.

In this review, we aim to cover important aspects related to the adaptation of animals to T_a_, the influence of T_a_ on the development of metabolic phenotypes in experimental mice, and their translation into human physiology.

## 2. Classification of Animals According to Body Temperature and Their Adaptation to Ambient Temperature

All living beings are sensitive to a minimum, optimum and maximum temperature. Due to environmental adaptations, organisms are conditioned to their habitat in different climatic zones. Accordingly, they are classified into eurytherms (tolerant to a wide variation of external temperatures) and stenotherms (tolerant to a narrow range of ambient temperatures) [18] (Figure 1).

The temperature of an animal is the amount of heat per unit of tissue mass and is a balance between heat production and exchange, a key determinant in reproduction and development [19]. Body temperature (T_b_) is defined as the reflection of the thermal energy that is retained in the body’s molecules. Based on the stability of T_b_, animal species can be classified as either poikilotherms or homeotherms [19] (Figure 1). Poikilotherms are animals with a variable T_b_, i.e., their temperature changes in response to environmental conditions. In contrast, homeotherms are animals that maintain a relatively stable T_b_. Most homeotherms manage to maintain a constant T_b_ through physiological processes that regulate production rates and heat loss. The difference between poikilotherms and homeotherms depends on the animal’s physiology and the nature of the environment. An animal can maintain a constant T_b_ if it inhabits an environment with a constant T_a_. Thermoregulation mechanisms are understood as the physiological strategy that an animal uses to control temperature within the desired range [20]. According to these thermoregulation mechanisms, animals are also described as ectotherms and endotherms (Figure 1).

In addition, animals can control their T_b_ through their behavior. Behavioral thermoregulation can be used to control the body temperature of a poikilotherm or to reduce the cost of thermoregulation in a homeotherm [20]. In ectotherms, the environment and behavioral thermoregulation determine the T_b_. In contrast, endotherms are vertebrates that generate internal heat to maintain a given T_b_.

Most mammals and birds (as illustrated in Figure 1) are classified as homeotherms because T_b_ is stable, and endotherms since they thermoregulate T_b_ through metabolic heat and the thermal insulation capacity of the animal.

## 3. The Use of Experimental Mice as a Model in Human Research

### 3.1. Relationship Between Body Size and Physiological Temperature

Since the 1990s, genetic mouse models have been used to study obesity and energy balance. The cloning and characterization of mutant genes associated with obesity led to the discovery of proteins such as leptin [21], its receptor [22] and melanocyte-stimulating hormone [23], among others [24,25,26], that cause monogenic obesity in mice and humans [27]. Together, these studies validated the use of the mouse in the modelling of biological diseases related to energy homeostasis. Nonetheless, in energy homeostasis studies, the thermal physiology of the experimental model of choice must be considered [8]. Mice and humans are both endothermic mammals with the ability to thermoregulate to maintain a constant T_b_. Yet the process of thermoregulation has important physiological differences between the two species that we should bear in mind as researchers. For instance, the size of an animal influences its thermal biology through its surface/volume ratio. The larger the individual is, the smaller the ratio. Body surface area is proportional to the power of 2/3 to mass and it is an important determinant in heat loss [28,29]. Adult humans are approximately 3000 times heavier than mice (75 kg vs. 25 g). As thermal biology depends on body size, it is important to consider this significant difference in inter-species dimensions. Homeothermic endotherms, such as mice and humans, must dissipate the excess heat produced by their metabolism across the body surface. Humans have a larger body size with a lower relative surface area, which leads to less heat loss. In contrast, mice have a smaller body size for a greater relative surface area, and thus greater heat loss.

Mice and humans have a similar internal T_b_ average of 37.0 °C in humans and 36.6 °C in mice [30], which is within the characteristic range in mammals. Humans generate heat primarily as a by-product of metabolism, without as much need for additional heat generation mechanisms. In fact, human physiology is mostly aimed at heat dissipation. In contrast, the small size of the mouse means that it can transfer heat quickly and have rapid changes in T_b_, so mice require more heat generation capacity to maintain their T_b_.

Figure 2 shows the components of energy expenditure depending on T_a_ [28,29,30,31]. In the mouse, total energy expenditure is the sum of the basal metabolic rate (BMR), physical activity, food thermogenesis and cold-induced thermogenesis [32]. At a given T_a_, over a third of the total energy expenditure is cold-induced thermogenesis, which is necessary to maintain T_b_. This amount of cold-induced thermogenesis, also called facultative or adaptive thermogenesis, is reduced by the availability of nesting material or by keeping mice grouped in cages so they can snuggle. In contrast, in humans, cold-induced thermogenesis contributes a very small fraction to total energy expenditure [33].

### 3.2. Thermal Physiology and Thermoneutrality Zone

Mammals use heat conservation and generation mechanisms to maintain thermal homeostasis, which is reflected in their constant internal temperature [20]. On exposure to a cold environment, several behavioral mechanisms of heat conservation are activated, such as vasoconstriction, piloerection, hunched posture (to reduce the surface area) and snuggling. When these conservative heat adaptations prove insufficient for defense against the cold, mammals increase their energy expenditure to generate heat by involuntary muscle contractions (shivering thermogenesis) and uncoupled respiration in brown adipocytes (non-shivering thermogenesis, known as adaptive or facultative thermogenesis). The opposite occurs when mammals face environmental heat. In this case, there are behavioral adaptations such as vasodilation and increased passive heat loss, as well as panting, licking and sweating (in humans) to increase active heat loss through cooling by evaporation.

Halfway between these metabolic adaptations to a cold environment and heat is the thermoneutral zone, which is defined as the nadir in BMR [8,9,10,20]. When the T_a_ is within the thermoneutral zone, BMR generates enough heat to maintain a constant core temperature at 37–38 °C. For young C57BL/6J mice (~ 3 months), the thermoneutral zone is between 29–31 °C [6,8,10,11], which is similar to the thermoneutral zone of a naked human (~28 °C) [34,35,36]. However, the thermoneutral or comfort zone in dressed humans is around 20–22 °C, which is often the temperature of the animal facilities where the mice are housed. This colder T_a_ keeps mice in significant thermal stress or controlled hypothermia, resulting in the activation of facultative thermogenesis in BAT to maintain thermal homeostasis. As a consequence, the BMR and food intake of mice housed at a T_a_ of 20 °C is ~100% higher than those housed at 30 °C. Both parameters increase by another ~100% when mice are housed at a T_a_ of 4–5 °C [37]. As discussed in detail below, the chronic housing of mice under thermal stress conditions (T_a_ of 20–22 °C) has profound effects on many physiological phenotypes and their intrinsic ability to adapt to environmental challenges.

Although the thermoneutral zone is considered a standard range, it is a highly variable parameter that differs between species. Previous studies showed that the thermoneutral zone of a particular mammal reflected its adaptations to its natural habitat [9].

In addition to these differences between species, many parameters can affect the range of the thermoneutral zone and the cold tolerance within a given species. For example, age (newborn and young mice have higher thermoneutral zones), muscle mass (basal metabolism and heat production are proportional to muscle mass), locomotor activity (exercise increases the production of heat to decrease the thermoneutral zone), pregnancy (fetal metabolism increases heat production), lactation (milk production generates heat), and isolation (greater isolation reduces the increase in metabolic rate at lower temperatures) can dynamically modulate the thermoneutral zone and the susceptibility of the organism to a cold environment [9,10,11]. This variation in the thermoneutral zone explains the differences observed in the cold tolerance of some mutant animals [11], such as those lacking hair, skin or subdermal fat [38,39,40,41,42]. This evidence suggests that experimental determination of thermoneutrality is necessary to understand how genetic mutations in mice affect physiology and disease susceptibility.

### 3.3. Thermal Variations in the Housing of Experimental Mice

In the animal facility, mice can consume unlimited food to meet the energy requirements of adaptive thermogenesis. However, it is known that “control” mice (wild-type experimental mice fed ad-libitum and without physical activity) become sedentary, obese and glucose intolerant and the implications for data misinterpretation in human studies is known [43]. The researchers stated that lack of exercise and unlimited access to food are the factors that most influence the inadequate interpretation of results. Other studies indicate that the underlying role of cold ambient temperatures is the cause of excessive intake and metabolic disorders [6,7,44]. A clearly defined stable environment is essential to generate consistent experimental results that support both replication and valid interpretations of the data. As animal models and measurement techniques become increasingly precise, environmental influences become critical in experimental development. Current technology can detect subtle effects that may have been part of the experimental background previously.

There are many varieties of rodent cages (for example, open lid, closed lid, ventilated and unventilated) that may vary in size, bedding, enrichment devices and other attributes. Even the position of the cage on a shelf can influence the result of the behavioral tests [7]. Another attribute that is rarely considered is the color of the cage. A few years ago, it was shown that this fundamental characteristic of the environment significantly influences circadian metabolic measures in rats [45,46]. The cage dye (transparent, amber, blue or red) causes a significant variation in maximum levels and maximum durations of melatonin during the dark phase and significant changes in the circadian moment of insulin spikes [46].

A fundamental characteristic of the rodent cage is the bedding. The properties of different types of rodent beds can differentially influence the environment of the cage, the physiology and behavior of rodents and even the health of the animals [47,48,49,50,51].

T_a_ is another critical feature of the rodent cage environment that is probably influenced by the type of cage system used. Some studies show how the T_a_ interacts with the cage system and possibly with tumor growth [12,52]. For example, one study evaluated the thermogenesis of BAT in nude and SCID mice that were individually housed at a T_a_ of 21 °C in ventilated cages with or without shelter or in a static (non-ventilated) cage. The results showed that, independently of the strain, mice individually housed in ventilated cages without shelter had significantly higher BAT thermogenesis and higher adrenal weights than mice housed in static cages or in ventilated cages with shelter. In addition, when tumor cells were implanted, mice housed in static cages had greater tumor growth than mice under the other two conditions. The authors concluded that mice housed in ventilated cages without shelter experienced cold stress, which in turn interfered with tumor growth [12]. Another study reported that BALB/c and C57BL/6 mice housed 5 per cage at a T_a_ of 22 °C had higher tumor growth than those maintained at 30 °C but did not detect a temperature effect on tumor growth when they used nude mice and SCID with immunodeficiency. The study also determined that the antitumor immune response was attenuated in immunodeficient mice maintained at 21 °C compared to those housed at 30 °C [52].

Another variable to consider is the density of mouse housing. For some studies, individual housing is preferred or necessary, while in other cases, rodents can be accommodated in groups that vary in number and density depending on the type of cage, the duration of the study, the purpose of the study and other factors. An important fact to consider is that not all mice housed in the same cage are necessarily identical, even if they are highly inbred. The differences between cage mates can be visually obvious in cages with domination hierarchies, which can occur in association with fights and overt injuries in some cage mates but not in others. Mice housed in groups may show greater phenotypic variation in some characteristics than mice housed individually from the same inbred strain [53].

Rodent housing density can directly affect the environmental conditions within the cage and, therefore, potentially alter the physiology, behavior of animals [54] and stress levels [55,56]. An interesting study evaluated the effect of the number of mice in a cage on the inside environment. Mice were housed in stable cohorts of one or five per cage, or in a test cage that initially contained five mice. One mouse was removed per week from the test cage until only one was left. Regardless of the room temperature (22 °C, 26 °C or 30 °C), cages containing five mice were general approximately 1.5 °C warmer than cages with individual mice, and a population of approximately three mice was associated with a decrease in temperatures and dew point inside the cage [57]. These findings are particularly relevant for situations in which individual mice are removed from a cage for some reason (for example, death, fighting and experimental use) because the remaining mice will experience different environmental conditions that could influence the experimental results [57].

Social housing can also affect the physiology and behavior of animals [58,59,60]. For example, a study that evaluated sleep, temperature and activity compared these measures in mice initially housed as part of a trio, then individually and finally individually with access to a shelter [60]. The data showed that the modifications in housing significantly influenced both the sleep and activity of mice. When housed individually, mice showed less rapid eye movement sleep and more locomotive activity during the dark phase than when they were housed as part of a trio. When given a shelter, the same mouse spent more time in slow wave sleep and was less active during the dark phase.

Thus, researchers should keep in mind that eliminating mice during an experiment could affect metabolism, as well as many of the other temperature-sensitive biological and physiological responses that have been analyzed so far. This probably contributes to experimental variability between experiments and laboratories.

## 4. Neuronal Control of Body Temperature

Homeostatic control of the T_b_ is essential for the survival of mammals. It is well-established that T_b_ is regulated partly by specific neuronal populations located in the hypothalamus [61]. This part of the brain works as a thermostat to maintain the T_b_ within a narrow range [62]. The most important regions of the hypothalamus involved in T_b_ regulation are the preoptic area (POA) and the posterior hypothalamic area (Figure 3) [63]. They contain temperature-sensitive neurons that initiate neuronal responses for heat generation or heat dissipation. This means that the brain itself is an input to regulate homeostatic responses. These conclusions are based on results obtained from electrophysiological recordings of the POA and revealed that local or environmental heat activate a subset of neurons referred to as “warm-sensitive” [8,64,65].

In addition to sensing local brain temperature, POA neurons receive thermal information from the periphery. It has been reported that three tissues provide an important input: the skin, spinal cord, and abdominal viscera [66]. The thermosensitivity of these tissues is due to sensory neurons that measure the temperature. Most of the neurons have cell bodies located in dorsal root ganglia (DRG) and their axons extend out to the target tissues [67]. Considerable progress has been made to elucidate the molecular basis of peripheral cold and warmth sensing. These studies have led to the identification of a number of ion channels activated by a wide spectrum of physical and chemical stimuli. Those activated by temperature belong to a superfamily of ion channels called transient receptor potential (TRP) channels [68,69]. Four TRP subtypes are activated by an increase in temperature and two TRP channels are activated by decreases in temperature [68]. For example, TRPM8 is an ion channel that admits Ca2^+^ and Na^+^ in response to moderate cold (10–25 °C), while several transient receptor potential cation channels (TRV) have been proposed to sense warmth including TRVP1, TRVP3, TRCP4 and TRVP2 [70]. The mechanism by which temperature modulates TRP channels remains to be elucidated.

Temperature information is sensed by these thermoreceptors in DRG neurons and is then transmitted to the dorsal horn of the spinal cord, where it is further processed before being sent to the brain (Figure 3). Elegant experiments have been carried out to elucidate the role of these thermosensitive neurons. Transgenic mice lacking TRVP1 in temperature-sensitive DRG neurons have reduced spinal neuron responses to heat [71]. Similarly, ablation of TRPM8+ DRG neurons reduced the number of spinal neurons activated by mild cold, but not by lower temperatures [72]. These results support the idea that spinal neurons synthesize information from many types of DRG neurons.

Dorsal horn neurons send glutamatergic projection to the brain that synapse with the lateral parabrachial nucleus (LPB) and the thalamus. Thermal information received in the thalamus is relayed upward to the somatosensory cortex and other cortex areas, where it mediates the discrimination of temperature (spinothalamocortical pathway) [73]. The ablation of this thermosensory pathway does not affect the autonomic response of T_b_ regulation. However, injuring or silencing of the LPB abolishes autonomic responses to skin cooling and warming and the temperature preference in behavioral assays [74]. This result suggests that the spinothalamocortical pathway does not play a role in the thermal afferent pathway that evokes involuntary thermoregulatory responses to environmental challenges.

Ascending temperature information terminates in two anatomically distinct areas of LPB: the external lateral and dorsal LPB (LPBel and LPBd). It has been demonstrated that warm and cold activate cFOS expression in LPBd and LPBel, respectively [75]. LPB neurons send glutamatergic projections to the midline POA, where GABAergic and glutamatergic interneurons in the median preoptic (MnPO) subnucleus are activated [76]. LPBel neurons activate GABAergic MnPO interneurons that inhibit the distinct population of warm-sensitive neurons in the medial preoptic (MPO) subnucleus that control cutaneous vasoconstriction, BAT and shivering. Thus, inhibition of neurons in the MPO increases core body temperature, shivering, metabolism and heart rate. In contrast, glutamatergic interneurons in the MnPO, which may be excited by glutamatergic inputs from warm-activated neurons in LPDd, excite warm sensitive neurons in MPO [61,77]. Altogether, this thermoregulatory network is a sophisticated reflex that is necessary to maintain T_b_ during an environmental temperature challenge (Figure 3).

## 5. Adaptive Thermogenesis in Brown Adipose Tissue

Small mammals have a tissue dedicated to heat generation, the BAT [78]. For a long time, it was known that BAT was present in small mammals such as rodents and neonatal humans. However, in the last decade it was discovered that active BAT is also found in adult humans [79].

In rodents, BAT is located mainly in the interscapular zone. In adult humans, it is found in the supraclavicular region, and in the cervical, axillary, paravertebral and perirenal areas [80]. BAT is called “classic” to distinguish it from inducible or beige adipose tissue, which has unique molecular and developmental characteristics [81]. Beige adipocytes have the appearance of white adipose tissue (WAT) until the animal needs to generate more heat. After exposure to cold or other stimuli, this beige adipose tissue or inducible BAT is enriched in cells with the appearance and functional characteristics of classic BAT in a process called browning. Although beige and BAT adipose tissue have different developmental origins and gene expression profiles [82], both are thermogenic. Thermogenic adipocytes can increase energy expenditure and generate heat by uncoupling the oxidative metabolism from ATP production. This function is carried out by the uncoupling protein (UCP)1, a proton transporter located in the internal mitochondrial membrane that uncouples energy generation from fuel oxidation from ATP production to produce heat. Thus, the electrochemical gradient generated through the electron transport chain (ETC) is dissipated [83,84,85]. In brown adipocytes, the high content of mitochondria and their vascular and nervous supply facilitates thermogenesis activated by the sympathetic nervous system. The nerve terminals act on α-adrenergic receptors to promote thermogenesis in BAT. It has been shown that cold enhances sympathetic signaling and that chronic exposure to cold triggers the expansion and activation of BAT [86], resulting in adaptive thermogenesis.

Adaptive thermogenesis is a mechanism of metabolic heat production that involves stimulation of the sympathetic nervous system to release norepinephrine (NE) and epinephrine, resulting in the increased metabolic activity necessary for heat generation in BAT [10,79]. Previous studies have shown that heat production by adaptive thermogenesis in mice can triple that of basal metabolism, and it is what increases the most in other animal models [87,88].

Obesity is an important risk factor for type 2 diabetes and cardiovascular disease. Importantly, BAT has been shown to promote HDL turnover and reverse cholesterol transport [89]. The high metabolic activity of thermogenic adipocytes confers atheroprotective properties through increased systemic cholesterol flow through the HDL compartment.

The thermogenic function of BAT requires an adaptive increase in proteasomal activity to ensure the quality control of cellular proteins. It has been shown that ER-localized transcription factor nuclear factor erythroid-2, like-1 (Nfe2l1 protein, also known as Nrf1) is an important mediator of brown adipocyte function, providing a greater proteometabolic quality control to adapt to cold or obesity [90]. It has been described that obesity might affect BAT’s proteasomal activity [90]. A recent epigenomic study associated an altered methylation pattern of the human NFE2L1 locus with BMI [91]. However, the molecular mechanism implicated in how this epigenetic variant could affect Nrf1 and proteasome activity is still unknown.

## 6. Therapeutic Efficacy of Adaptive Thermogenesis in Obesity

In recent decades, BAT has been extensively investigated for its potential therapeutic role in obesity and T2D. Previous studies showed that excessive caloric intake could stimulate the expansion of BAT and the increase in thermogenesis as an adaptive measure to maintain body weight. This mechanism of diet-induced thermogenesis is mediated by BAT and UCP1 [92]. In fact, in the absence of UCP1, mice are prone to obesity. Initial studies, where brown adipocytes were genetically ablated with a toxin driven by the UCP1 promoter [93], demonstrated for the first time the protective effect of BAT against obesity and T2D. Importantly, these protective effects were observed in mice raised at room temperature (thermal stress with a T_a_ of 20–22 °C). Subsequent investigations under T_a_ conditions showed that *Ucp1^−/−^* mice were very susceptible to hypothermia, due to recurrent tremors, but did not demonstrate the role of UCP1 in thermogenesis, nor a propensity to develop obesity [94,95]. However, when mice remained at thermoneutrality, they showed greater metabolic efficiency, which resulted in an increase in adiposity and obesity development [96]. This is explained because UCP1 knockout mice are more susceptible to hypothermia, which directly affects most of the systemic effects of energy metabolism [93].

The preferable fuel source in BAT is lipids, but glucose is also used. Therefore, approaches to activate BAT and reduce glucose and lipid content through adaptive thermogenesis could be potential therapies to fight against obesity [83]. The best way to model human energy physiology with mice is under thermoneutral (30 °C) conditions. Under this situation, cold-induced thermogenesis would be minimal and will not influence total energy expenditure [97]. Other potential variables that may account in similar proportions for total energy expenditure compared with a sedentary human would be BMR (70%), food thermogenesis (10%) and energy expenditure by physical activity (20%) [7]. Because energy expenditure decreases by approximately 50% in mice at thermoneutrality [98], this implies that the metabolic phenotype in obesity including adiposity would be highly dependent on the T_a_. An example of this implication is shown in a study about thyroid hormone metabolism [99]. In humans, hyperthyroidism is associated with a hypermetabolic state, characterized by heat intolerance and fat loss, while hypothyroidism decreases energy expenditure and promotes cold intolerance and obesity. Interestingly, the authors have shown that unlike in humans with hypothyroidism, mice that lack type 2 deiodinase, a key enzyme in the conversion of thyroid hormone, did not develop metabolic dysfunction when housed at 22 °C. However, when these animals were maintained at thermoneutrality (30 °C), there was an increase in adiposity, hepatic steatosis and glucose intolerance [99]. Therefore, they concluded that the accommodation of mice at a T_a_ resulted in increased adrenergic activity in BAT, which compensated for the loss of activity of deiodinase type 2 and the production of T3. These findings suggest that chronic housing of mice under conditions of thermal stress can mask the genetic functions involved in energy balance and metabolic homeostasis triggering a change in the metabolic phenotype.

### 6.1. Activating BAT to Treat Obesity

Obesity is a chronic metabolic disorder characterized by ectopic fat deposition and a state of chronic low-grade inflammation. It is associated with higher free fatty acids, glucose and insulin levels. Adipose tissues, both WAT and BAT, are highly affected during obesity. These alterations include adipocyte hyperplasia and hypertrophy [100], endoplasmic reticulum stress [101], oxidative stress [102], fibrosis [100] and mitochondrial dysfunction [103], among others. Obesity is associated with severe disorders such as cardiovascular disease, dyslipidemia, T2D or even some forms of cancer. BAT was initially recognized for its ability to protect animals from hypothermia [104]. However in the last decade, the discovery that BAT is active in adult humans and that it is reduced in several conditions such as obesity, T2D and aging has triggered leading research in the BAT field to improve lipid and glucose homeostasis in the fight against obesity [105,106,107,108,109].

Overfeeding activates BAT’s diet-induced thermogenesis [92]. Thus, several studies have focused on natural or chemical drugs to enhance thermogenesis such as ginger [110], tea seed oil [111], berberine [112], butein [113], capsaicin [114], flavonoids such as quercetin [115], calcium supplement [116] or fluvastatin sodium [117], among others.

Exercise is another inductor of BAT thermogenesis [118]. Interestingly, a positive correlation has been demonstrated between exercise and increased browning in subcutaneous WAT [119]. The lactate produced in muscles after exercise or after cold exposure leads to an increase in UCP1 levels in the adipose tissue [120]. Recently, Takahashi et al. demonstrated that subcutaneous WAT-derived TGF-β2 secreted after exercise or its administration improved glucose homeostasis and insulin sensitivity by increasing fatty acid oxidation [121]. Furthermore, some myokines related to BAT activation, such as irisin [122] or β-aminoisobutyric acid, have been shown to decrease weight gain and improve glucose tolerance in mice [123].

As mentioned above, BAT is innervated by the sympathetic nervous system and controlled by adrenergic inductors such as norepinephrine. Thus, huge efforts have been made to find novel β-adrenergic agonists that can potentiate BAT activity and enhance thermogenesis. Some examples are Cl-316,243 [124] or mirabegron [125]. The latter was initially clinically used for overactive bladder but was also found to activate BAT in rats and humans [126]. Many other studies have focused on increasing the BAT mass, i.e., the differentiation of brown adipocytes or WAT browning. One study centered on fibroblast growth factor-21 (FGF21) [127], which is mainly secreted by the liver and associated with BAT activity. In humans higher FGF21 levels have been found in serum after cold exposure [128]. Another important thermogenic coactivator is PGC1-α, which is involved in mitochondrial biogenesis and thermogenesis [129]. Its activity is related to an increase in other transcriptional factors involved in brown differentiation such as PRDM16 or the PPARs family [130]. Other factors have also been studied to enhance thermogenesis in obesity such as the bone morphogenetic proteins (BMPs) family, for example BMP7 [131] or BMP8b [132]. Interestingly, although BMP4 improves the obese phenotype, it has a tissue-dependent dual effect: it increases browning in subcutaneous WAT; and it increases the number of lipid droplets and decreases BAT UCP1 expression [133]. Another approach to activate thermogenesis has been based on PPAR-γ agonists. Some studies showed increased UCP1 levels after treatment with rosiglitazone [134,135]. Other secreted peptides or hormones have been reported to activate BAT: norepinephrine [136], natriuretic peptides [137], meteorin-like [138], bile acids, adenosine [139] or activin E [140,141].

In recent years, BAT-derived adipokines, commonly called batokines, have generated considerable interest among the scientific community for their anti-obesity potential [142]. In 2018, Deshmukh et al. found a batokine called EPDR1 involved in BAT activation [143]. Another recent and elegant study has discovered a new chemokine called CXCL14 that is secreted by BAT and induces browning of WAT via immune cell activation [144].

Finally, alternative therapies are being studied to increase BAT mass and thermogenesis. These include the direct transplantation of this tissue or differentiated beige cells from preadipocytes, mesenchymal stem cells (MSCs) or induced pluripotent stem cells (iPSC) [145]. In the future, more personalized therapies could focus on the intrinsic study of the genome to identify other BAT activators such as miRNAs to combat obesity [146].

### 6.2. Role of Thermoneutrality in Obesity and Metabolic Studies: Chronic Cold vs. Thermoneutrality

It is known that an increase in metabolic heat production has physiological effects. In fact, for every 1 °C that the T_a_ drops, approximately 46.3 kcal/m^2^/24 h are required to maintain the core temperature of the mouse [13]. This increase in metabolic heat production has many physiological effects. For example, energy expenditure is approximately 50% lower in mice living at thermoneutrality than in mice living in chronic cold conditions [98]. Therefore, we can deduce that the metabolic phenotypes of obesity and adiposity are highly dependent on the T_a_ at which mice are housed (Figure 4).

Some studies have shown that the lack of efficient induction of the obese phenotype by high fat diet (HFD) is due to the fact that mice were housed in conditions below their thermoneutrality [57,58,59,60]. This effect of low temperatures on obesity has been observed in other animal models [14,78,81], demonstrating the correlation between low temperature and low effectiveness of the diet, as well as low levels of insulin and glucose and an altered response in glucose tolerance tests and energy homeostasis [10,44].

Other researchers found differences in metabolic inflammation [84]. Mice housed at thermoneutrality vs. chronic cold had greater metabolic inflammation. This was correlated with higher inflammation in the WAT and at vascular level, promoting atherosclerosis. The authors thus claim that thermal stress could limit our ability to faithfully model human diseases in mice [84].

The exposure to chronic cold vs. thermoneutrality has also been studied at the pharmacological level in obesity. The chemical protonophore, 2,4-dinitrophenol (DNP) was used for weight loss in humans. However, its consumption was discontinued due to toxicity [45,46]. DNP generates heat by uncoupling mitochondria. Thus, its anti-obesity mechanism is at least due to an increase in energy expenditure, without a direct effect on food intake. At 22 °C, chronic treatment in mice with a low dose of DNP had no effect on the phenotype and the only change observed was that BAT became less active [47]. This indicates that adaptive cold-induced thermogenesis was potentially reduced by the amount of heat that was generated by DNP-mediated uncoupling [147]. When the same experiment was performed at thermoneutrality, the same dose of DNP increased energy expenditure, reduced body weight, reduced adiposity and improved glucose tolerance [47]. Although DNPs have been discontinued for the treatment of obesity [46], they could serve as a model for the effects of systemic uncoupling agents, which demonstrate better efficacy at thermoneutrality than at colder temperatures.

Another comparison of the pharmacological effect at chronic cold vs. thermoneutrality is the β3 adrenergic agonist, Cl316,243. The main effects of β3 agonists are direct stimulation of BAT thermogenesis and WAT lipolysis [52]. These experiments were performed at 22 °C, and the pharmacological treatment showed no effect on body weight or adiposity. However, there was greater energy expenditure compensated with the increase in food intake [52]. At thermoneutrality, Cl316,243 increased energy expenditure, but it also reduced body weight and adiposity.

Altogether, it is clear that the T_a_ used in experiments with mice is critical and might directly influence the efficiency and clinical translatability of pharmacological, metabolic and energy homeostasis studies.

## 7. Conclusions and Future Perspectives

The mouse is the predominant model for studying human diseases. However, many studies fail to deliver mechanistic information about human physiology. This failure comes from translating preclinical studies in mice to therapy in humans. Although we are aware that mice and humans are two different species, we do not always consider all the external variables of the environment, which could influence the physiological adaptation of the study model and hinder the reproducibility of preclinical investigations.

The thermoregulatory network triggered by the hypothalamus is a necessary reflex to maintain T_b_ during variations in T_a_. Therefore, given the influence that T_a_ exerts on the physiological and pathophysiological responses of the mouse, this study variable should be considered for more correct, efficient translation into human therapy.

The epidemic of obesity and metabolic diseases is increasing exponentially, and current therapies remain inefficient. BAT has become an attractive potential target to treat obesity due to its thermogenic capacity. However, environmental variables such as temperature could directly influence both BAT activity and the dynamics of energy expenditure in the model under study.

For future metabolic studies, it would be important to consider all the variables that may influence the experimental outcome regarding obesity and insulin resistance. For example, an important point is the selective breeding of mouse animal models. Relevant specific differences in metabolic activity have been found in strain-dependent genetic conditions [148,149,150]. However, the correlation between strain variables and thermo-neutrality in studies of obesity and associated diseases is still unknown.

Currently, most metabolic studies are carried out at a room temperature of 21 °C, which is considered a thermoneutral zone for adult humans. However, mice subjected to the same temperatures experience chronic cold. The cold triggers-controlled hypothermia in which energy expenditure is affected by the increase in adaptive thermogenesis, by the activation of BAT and tremors due to involuntary muscle contractions (Figure 4). Further studies in mice at thermoneutrality will deepen our understanding of physiological mechanisms, which could increase the success of translation into human treatments for obesity and metabolic diseases.

## Figures and Tables

**Figure 1 cells-09-00316-f001:**
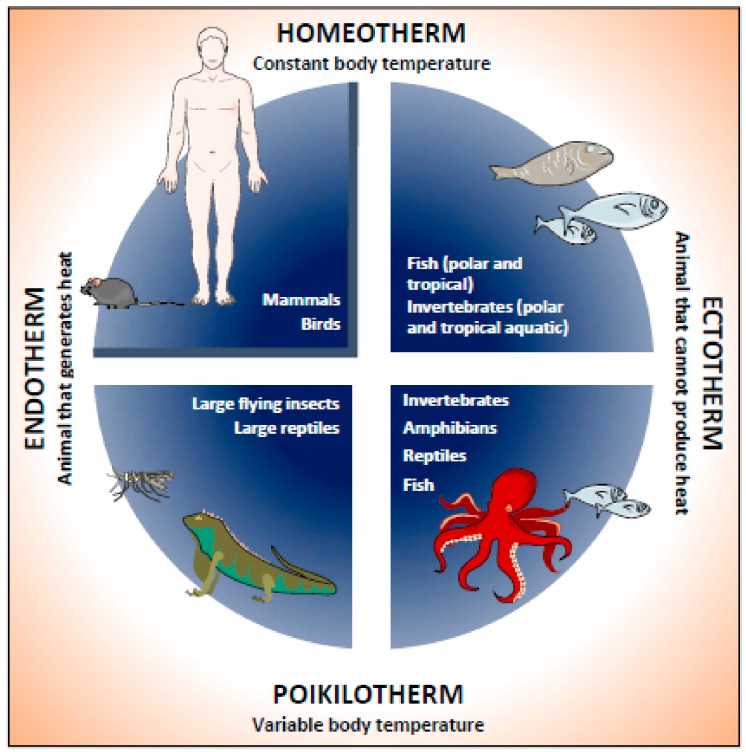
Classification of animals according to their body temperature and adaptation to room temperature. Animals are classified according to their way of acquiring body heat and their ability to adapt to room temperature. Endothermic animals can produce heat endogenously. Ectothermal animals cannot produce their own heat, so they rely on ambient temperature. In addition, animals can be classified into homeotherms that can keep their body temperature constant, or poikilotherms that have a variable body temperature. Humans and mice are mammals that are classified as homeothermal endotherms. However, when subjected to an ambient temperature well below their thermoneutrality, they experience physiological responses that trigger adaptive thermogenesis.

**Figure 2 cells-09-00316-f002:**
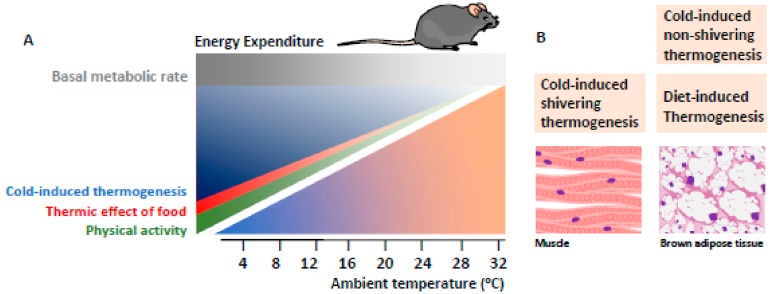
Cellular energy used in adaptive thermogenesis. (**A**) The total energy expenditure of the mouse can be divided into four components: basal metabolic rate, physical activity (green), the thermal effect of food (red) and cold-induced thermogenesis (blue). At room temperature (20–22 °C) more than one third of the total energy expenditure is cold-induced thermogenesis, which is required to maintain body temperature. (**B**) The energy used for cold-induced thermogenesis is mainly produced by skeletal muscle shivering and BAT thermogenesis.

**Figure 3 cells-09-00316-f003:**
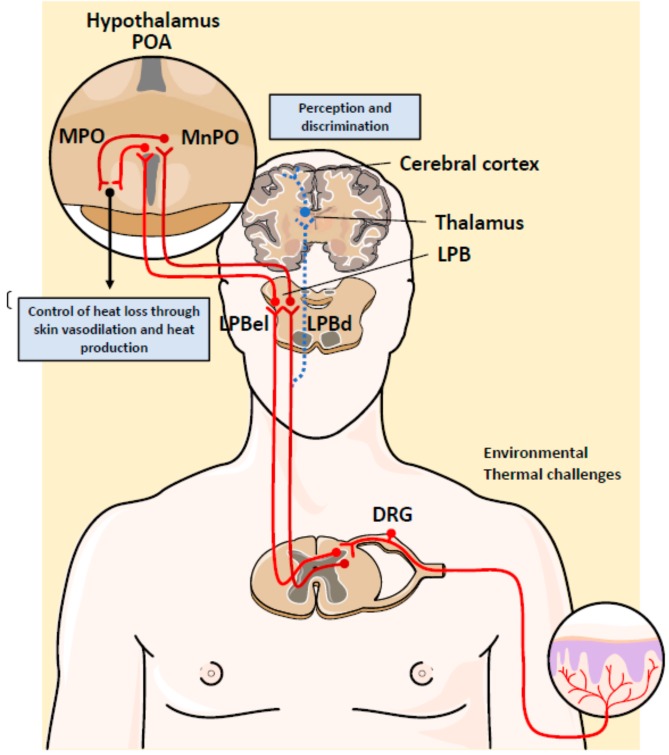
Hypothalamic thermoregulatory network that controls body temperature in mammals. The hypothalamic neurons of the preoptic area (POA) and the posterior hypothalamic area function as a body thermostat to maintain a stable body temperature. The ascending temperature information ends in two anatomically distinct areas of the lateral parabrachial nucleus (LPB): the lateral and dorsal external LPB (LPBel and LPBd, respectively). Heat and cold have been shown to activate cFOS expression in LPBd and LPBel, respectively. Altogether, this sophisticated thermoregulatory network highlights the importance of maintaining body temperature during an environmental temperature challenge. DRG, dorsal root ganglia; MPO, medial preoptic subnucleus; MnPO, median preoptic subnucleus.

**Figure 4 cells-09-00316-f004:**
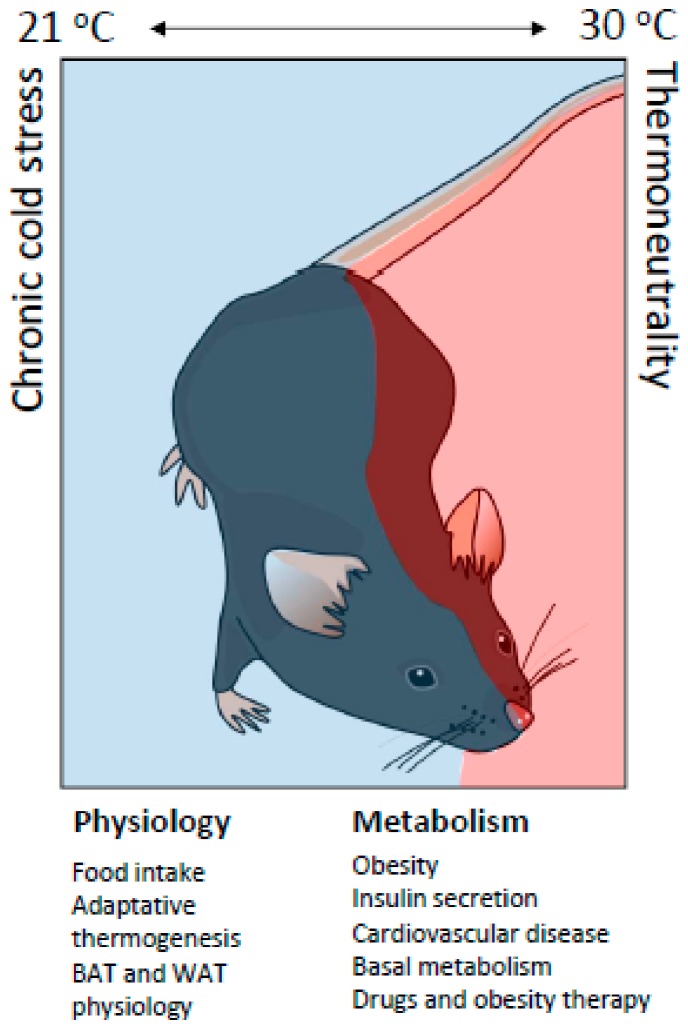
Metabolic and physiological differences in mice under chronic cold or at thermoneutrality. Mouse models used to study metabolic diseases are influenced by environmental temperature. Mice show differences in the metabolic phenotype when housed at a standard temperature (21 °C) vs. thermoneutrality (30 °C). Mice at standard temperatures are subjected to chronic cold. This triggers controlled hypothermia where energy expenditure is affected by changes in physiology (food intake, BAT and WAT physiology and an increase in adaptive thermogenesis) and in metabolism (basal metabolism, adaptive thermogenesis, diet efficiency, insulin secretion, adipose tissue physiology, inflammation at adipocyte and vascular levels, and the effect of drugs and therapies against obesity).

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
