# Peer review of "Impact of Adaptive Thermogenesis in Mice on the Treatment of Obesity"

_cells, 2020, doi:10.3390/cells9020316_

Round 1
Reviewer 1 Report
This is a comprehensive review and I like the figures. Two aspects that is not well covered are
Obesity is so dangerous because of the strong risk for diabetes and cardiovascular disease. The capacity of BAT to improve dyslipidemia, high atherogenic cholesterol and low HDL is not touched upon. There are some new mechanisms how brown fat fails in obesity, including the discovery of the Nrf1/Nfe2l1-proteasome pathway. This could be updated.
Author Response
We do agree with the Reviewer on the importance of including these two key aspects. Thus, we have added the following sentences in the revised ms (lines 352-363):
“Obesity is an important risk factor for type 2 diabetes and cardiovascular disease. Importantly, BAT has been shown to promote HDL turnover and reverse cholesterol transport1. The high metabolic activity of thermogenic adipocytes confers atheroprotective properties through increased systemic cholesterol flow through the HDL compartment.
The thermogenic function of BAT requires an adaptive increase in proteasomal activity to ensure the quality control of cellular proteins. It has been shown that ER-localized transcription factor nuclear factor erythroid-2, like-1 (Nfe2l1 protein, also known as Nrf1) is an important mediator of brown adipocyte function, providing a greater proteometabolic quality control to adapt to cold or obesity2. It has been described that obesity might affect BAT’s proteasomal activity2. A recent epigenomic study associated an altered methylation pattern of the human NFE2L1 locus with BMI3. However, the molecular mechanism implicated in how this epigenetic variant could affect Nrf1 and proteasome activity is still unknown”.
Bartelt A, John C, Schaltenberg N, et al. Thermogenic adipocytes promote HDL turnover and reverse cholesterol transport. Nat Commun. 2017;8:1-10. doi:10.1038/ncomms15010 Bartelt A, Widenmaier SB, Schlein C, et al. Brown adipose tissue thermogenic adaptation requires Nrf1- mediated proteasomal activity. Nat Med. 2018;24(3):292-303. doi:10.1038/nm.4481.Brown Wahl S, Drong A, Lehne B, et al. Epigenome-wide association study of body mass index , and the adverse outcomes of adiposity. Nature. 2017;541(7635):81-86. doi:10.1038/nature20784.Epigenome-wide
Reviewer 2 Report
This is a thorough, well-written review on topic of thermal regulation in mice, with a focus on implications for obesity research. The text is well-written. The topic is covered completely. I have only one suggestion: the authors might consider addressing strain specific differences in mice with respect to susceptibility or lack thereof to obesity and insulin resistance at lower temperatures would be helpful. Related to this, selective breeding has been used as tool to select for mice within a species that are more or less susceptible to obesity and insulin resistance at lower temperatures. Coverage of these issues would be of practical utility for investigators.
Author Response
The suggestion made by the Reviewer is excellent. Following his /her suggestions, we have included the following sentences in the revised version of the ms (lines 512-517):
“For future metabolic studies it would be important to consider all the variables that may influence the experimental outcome regarding obesity and insulin resistance. For example, an important point is the selective breeding of mouse animal models. Relevant specific differences in metabolic activity have been found in strain-dependent genetic conditions4,5,6. However, the correlation between strain variables and thermo-neutrality in studies of obesity and associated diseases is still unknown.”
West DB, Boozer CN, Moody DL, Atkinson RL. Dietary obesity in nine inbred mouse strains. Am J Physiol - Regul Integr Comp Physiol. 1992;262(6 31-6). doi:10.1152/ajpregu.1992.262.6.r1025 Montgomery MK, Hallahan NL, Brown SH, et al. Mouse strain-dependent variation in obesity and glucose homeostasis in response to high-fat feeding. Diabetologia. 2013;56(5):1129-1139. doi:10.1007/s00125-013-2846-8 Kodela E, Moysidou M, Karaliota S, et al. Strain-specific differences in the effects of lymphocytes on the development of insulin resistance and obesity in mice. Comp Med. 2018;68(1):15-24.